# The Road to Bring FDCA and PEF to the Market

**DOI:** 10.3390/polym14050943

**Published:** 2022-02-26

**Authors:** Ed de Jong, Hendrikus (Roy) A. Visser, Ana Sousa Dias, Clare Harvey, Gert-Jan M. Gruter

**Affiliations:** 1Avantium N.V., Zekeringstraat 29, 1014 BV Amsterdam, The Netherlands; roy.visser@avantium.com (H.A.V.); ana.sousadias@avantium.com (A.S.D.); clare.harvey@avantium.com (C.H.); gert-jan.gruter@avantium.com (G.-J.M.G.); 2Industrial Sustainable Chemistry, Universiteit van Amsterdam, Science Park 904, 1098 XH Amsterdam, The Netherlands

**Keywords:** FDCA, PEF, polyester, multilayer bottles, barrier properties, recyclability, film, fibre, biobased

## Abstract

Biobased polymers and materials are desperately needed to replace fossil-based materials in the world’s transition to a more sustainable lifestyle. In this article, Avantium describes the path from invention towards commercialization of their YXY^®^ plants-to-plastics Technology, which catalytically converts plant-based sugars into FDCA—the chemical building block for PEF (polyethylene furanoate). PEF is a plant-based, highly recyclable plastic, with superior performance properties compared to today’s widely used petroleum-based packaging materials. The myriad of topics that must be addressed in the process of bringing a new monomer and polymer to market are discussed, including process development and application development, regulatory requirements, IP protection, commercial partnerships, by-product valorisation, life cycle assessment (LCA), recyclability and circular economy fit, and end-of-life. Advice is provided for others considering embarking on a similar journey, as well as an outlook on the next, exciting steps towards large-scale production of FDCA and PEF at Avantium’s Flagship Plant and beyond.

## 1. Avantium’s Goal: Bringing Disruptive Technologies to the Market

Avantium is a Dutch technology company in renewable chemistry whose origins in advanced catalysis products and services laid the foundations for its growth into developing and commercialising sustainable chemical technologies. Headquartered in Amsterdam, Avantium currently employs approximately 220 people of around 20 nationalities, and has extensive R&D laboratories and operates three pilot plants: one in Geleen and two in Delfzijl, the Netherlands. Avantium’s foundation in high-throughput catalyst testing systems has been fundamental for the R&D approach used in their development programs using the benefits of parallel testing to speed-up the development time-lines [1,2,3,4]. This approach has allowed Avantium to make breakthroughs in the production of building blocks (monomers) from renewable sources (plant-based sugars and CO_2_) instead of from fossil resources such as petroleum. Plant-based monomers are urgently needed to tackle the various climate and material challenges currently faced by the world. This paper describes the history of the development of Avantium’s YXY^®^ Technology, and special emphasis is given to all the requirements (strong sustainable technology portfolio, production at pilot plant scale, IP protection, regulatory approval, commercialization strategy, recycling options, and fate in nature) needed to successfully bring a new polymer with bulk potential to the market.

### 1.1. Avantium’s Coherent Portfolio of Technologies

With a mission to move towards a fossil-free world, Avantium has developed multiple game-changing technologies at various stages of commercialization (Figure 1). Following the development of its YXY^®^ Technology, Avantium looked at the supply chain required to ensure that PEF can be produced in a responsible and sustainable manner, and initiated R&D programs where gaps were seen, resulting in the development of both the mono-ethylene glycol Ray Technology™ and the biorefinery Dawn Technology™ for second generation (2G) glucose. The combination of these technologies leaves Avantium poised to disrupt the plastic packaging material industry, covering the value chain from plant-based feedstock towards multiple end-applications such as packaging (e.g., bottles, trays and pouches), textiles, film and more everyday items. These growing end-markets equate up to a size worth over $200 billion per year.

Avantium’s most advanced technology is the YXY^®^ Technology that catalytically converts plant-based sugars into FDCA (furan dicarboxylic acid), the main building block of PEF: a 100% plant-based, fully recyclable plastic material with significant performance benefits and with a significantly lower carbon footprint than fossil-based plastics [5,6,7,8]. Avantium has successfully demonstrated the YXY^®^ Technology at a pilot plant in Geleen, The Netherlands. The second technology pursued is the Ray Technology™ that catalytically converts sugars in one step into glycols, such as plantMEG™ (mono-ethylene glycol), an important monomer for both PEF and PET and plantMPG^TM^ (mono-1,2-propylene glycol), among others, a building block for unsaturated polyesters as well as a functional fluid (de-icing, heat transfer). Avantium has a pilot plant to produce plantMEG™ and plantMPG^TM^ in Delfzijl, the Netherlands (Figure 2) using a highly efficient one-step hydrogenolysis process. The third technology is the Dawn Technology™ that converts non-food lignocellulosic biomass into industrial sugars and lignin in order to transition the chemicals and materials industries to non-fossil, non-food resources. This use of non-food biomass (so-called 2nd generation feedstocks) is an important future perspective to grow the industry in a sustainable manner [9,10,11]. Avantium runs a Dawn pilot biorefinery in Delfzijl, the Netherlands. In addition to Avantium’s technologies using plant-based carbon sources, Avantium aspires to also develop materials using carbon dioxide (CO_2_) as a feedstock. While for energy, many renewable options are available, for materials next to biomass, CO_2_ is the only other renewable carbon source. Avantium’s Volta Technology, an important technological step within carbon capture and utilization (CCU), is an electrocatalytic platform that converts CO_2_ into chemical building blocks and high-value products such as the polyester monomers oxalic acid and glycolic acid [12,13,14,15]. Finally, Avantium is applying its new plant-based and CO_2_-based building block opportunities by developing and evaluating novel and improved polymer products for the plastic materials of the future. The perspective of using electrochemistry in furan chemistry has also been researched and reviewed [16,17,18,19].

### 1.2. Production of Demonstrators to Engage Partners at an Early Stage

Avantium took the approach by setting out to use their catalytic process expertise and testing capabilities to convert sugars into a family of compounds called “furanics.” In 2004, the US Department of Energy issued an influential report called “Top Value-added Chemicals from Biomass” [20]. The 12 molecules with the greatest anticipated potential were evaluated with respect to economic and market potential. Avantium extended the list and developed their own (stage-gated) evaluation, realizing that production costs would be critical, especially for those chemicals already on the market, the so-called drop-in molecules. These molecules, such as biobased mono-ethylene glycol (MEG) and 1,4-butanediol, are fundamentally the same molecules as their fossil counterparts, with identical performance regardless of their origins. For decades, the chemical industry has been narrowly focused on increasing production scale, yields and reducing costs. This makes it extremely difficult to commercialize the same product from a different feedstock, as it will have to compete on cost from day one because partners in the value chain are only committed to a very limited “green premium”. Many sustainable chemistry companies have failed to recognize this challenge in time and had no plan on how to overcome this gap, leading to failure of the business [21].

It is different for new chemicals not yet on the market. These new molecules with unique functionality have the advantage of no fossil analogue to compete with. However, these new molecules and the resulting products will always be compared to existing comparable products. It is critical to investigate at an early stage any advantageous properties they may have that bring value to specific product-market applications. There can also be a greater tolerance to price if you are solving an unmet need. The key challenge for the development of any novel chemical product and/or process technology is that in the early days of commercialization, production volumes will be relatively small compared to incumbent, commodity chemicals. These relatively small plants still require multiple synthesis steps and require significant capital investments (Capex) and are in general less optimized for co-product valorisation, closed process streams as well as heat integration/energy recovery resulting in also higher operating expenditures (Opex).

As part of a strategic investment to accelerate process and application development in FDCA-based chemicals and plastics, Avantium signed a collaboration agreement with Cargill subsidiary NatureWorks to develop FDCA-based polyesters. This partnership came at an excellent time. Around 2008 Avantium was producing FDCA at kilogram scale and NatureWorks was one of the few companies which had commercialized a novel (biobased) polymer in decades (polylactic acid (PLA)) and therefore understood the process of matching polymer properties with end-use requirements. One of the newly investigated polyesters was PolyEthylene 2,5-Furanoate (PEF), which turned out to be a standout candidate as it had excellent barrier properties along with enhanced thermal and mechanical properties compared to PET. A strong barrier preventing oxygen entering the bottle (relevant for juice and beer) and CO_2_ leaving the bottle (relevant for any carbonated beverage) was an unmet need in the market. The industry was using special coatings, additives (scavengers), or nylon layers for PET bottles to improve this barrier. This not only added cost but also caused a challenge for the PET recyclers. PEF clearly has potential for solving this unmet market need. It was decided that potential partners in the value chain, e.g., converters, brand-owners, retailers and consumers, are more likely to be convinced by showing an actual product, a demonstrator. To make this possible, partners were selected who could make at 10′s of kg scale polymer of the required specifications, convert it into preforms, blow bottles and test the quality of the bottles (barrier, strength, colour, etc.). This resulted in the production of the so-called “golden” bottle (Figure 3). Luck and good timing can play an important role in innovation. At a climate convention in Copenhagen in 2009, our new PEF plastic bottle, partially made from plant-based sugars, was launched. Brand owners using bottles for water and carbonated soft drinks (CSD) were looking for 100% plant-based materials as an alternative for their petroleum-based PET bottle. A meeting with one of these CSD brand owners was arranged during which one of our first golden PEF bottles was placed on the table. It still had this “golden” colour (a consequence of insufficient monomer purity and initial polymerization conditions), but its list of properties sounded too good to be true. A material transfer agreement was signed allowing this company to independently evaluate PEF’s properties. When its outstanding properties were confirmed, the first multiyear joint development agreement (JDA) was signed with Coca-Cola to further develop PEF for bottles. At the same time, a similar JDA was signed with Danone and later ALPLA (a major PET converter from Austria) joined as a third partner. One of the first things that our partners requested was to stop calling this the “Furanics Technology” as ideally a new name should have no meaning (and certainly should not be anything chemical). Subsequently, Avantium introduced the name YXY for their sugar-based FDCA and PEF technology.

The “bottle consortium” partners were valuable as they gave us access to regulatory expertise and guidance on how to obtain food contact approval for PEF and toxicity testing of the FDCA monomer. ALPLA facilities for and experience with preform injection and bottle blowing as well as its background in PET recycling have been a strong support in the PEF technology development on these topics. The partnerships not only advanced the technical and regulatory development but were also instrumental in raising the money to complete two pilot plants for the key FDCA production steps (sugar dehydration and oxidation) in the Netherlands.

### 1.3. Production at Pilot Plant Scale

Since 2011, Avantium has successfully operated a YXY pilot-scale plant in Geleen (the Netherlands). The pilot plant is operated 24/7 all year round and has been operated to further develop and demonstrate the scalability of the YXY^®^ Technology and to produce sufficient volumes of FDCA and PEF to develop applications with partners. The purpose of a pilot plant is in general two-fold: (1) Proof the Process and (2) Proof the Product. For Proof the Process, it is very important that the unit operations in the pilot plant can be scaled and resemble the scaled-up operation. The Proof of Product is especially relevant in the case of the production of a novel building block/novel material. For the market development of FDCA and PEF, it is crucial that a pilot plant can produce enough material to validate the developed applications under real conditions, the so-called pre-marketing studies. With successfully passing the pilot plant goals, Avantium has commenced proceeding to the Flagship Plant stage, where the company begins production at commercial scale. For the Flagship Plant the PDP (Process Design Package) and FEED (Front End Engineering and Design) for the Flagship Plant have been finalized and the EPC-phase (Engineering, Procurement and Construction) has started. The data accumulated in the pilot plant were instrumental in achieving a successful PDP as well as FEED. Once the Flagship Plant is operational and the technology commercially validated, Avantium wishes to license its YXY^®^ Technology to industrial partners for broader scale deployment and market adoption.

### 1.4. The Importance of Patents and Patent Protection

IP rights are some of the most valuable assets of Avantium. Its technologies are covered by 137 patent families (2020) and are listed under different entities including Avantium, Avantium Knowledge Centre, Furanix and Synvina. Patents are a national (exclusive) right and parties can limit competitors to operate in a country if they have been granted a patent in such country. Protection of IP rights is a critical factor for Avantium’s selected business model of licensing. It has developed an extensive IP position that it continuously maintains and expands by filing new relevant applications in order to protect its proprietary technologies and products. Patent approvals in the high volume and performance markets Europe, the US and Asia are important, since their approvals are seen as leading in this field. In addition, specific patents are typically filed where they are relevant: bio feedstock conversion technologies are filed in countries with abundant biomass, and patents involving production technology are filed in countries which have an industrial infrastructure where (purified) terephthalic acid (PTA) or PET is produced. Basic end-use application patents are filed where those products are produced or consumed, however the main downstream development strategy for Avantium is to create an open innovation platform for FDCA and PEF applications [22,23,24,25,26,27,28,29]. Avantium is closely collaborating with its partners and customers with the goal to transfer and develop the knowhow to make the products work in their applications.

For its PEF technology, Avantium has a solid protection of its leading technology with 57 patent families representing more than 400 patent rights and covering the full PEF value chain. This is not only required to safeguard Avantium’s leading position in the production of FDCA and PEF but is also a prerequisite for the technology to be licensed out. Avantium has patents to produce FDCA precursors and FDCA, as well as to polymerise FDCA to PEF. The company has also patent protected many applications of PEF in a broader sense, such as bottles, fibres and films, as well as (chemical) recycling of PEF. In addition, Avantium also owns several patents on side streams of the process such as humins (10 granted, 16 pending) and on polymers other than PEF. Avantium has an active IP management program, with routine searches for current awareness, discussions with technicians to harvest additional inventions, protecting its position via patent oppositions and is active in licensing discussions. These activities are an essential part of the day-to-day business. The fact that other companies are also striving to produce FDCA demonstrates that the production of FDCA is a major market opportunity with large market potential in different application fields.

### 1.5. Chemical Registration and Food Contact Approval

The chemical registration of new substances is a critical step to allow chemicals to be produced and/or imported in the different jurisdictions. The rules for registration are different from region to region, nevertheless the basis of the registration allows the risk assessment of the impact of the (new) substances on human health and the environment. Depending on the region the polymers may be exempted of registration, either by the fact that the monomers are registered or because the polymer is considered a polymer of low concern. As an example, a brief summary of the registration of chemicals and polymers in European Union (EU) is given underneath.
Chemical registration in EU: Chemicals need to be registered according to the European Regulation often abbreviated by REACH (Registration, Evaluation, Authorization and Restriction of Chemicals). REACH requires the registration of the products produced and/or imported to the European Union including isolated intermediates when quantities are above 1 t per year, independently of the use. Depending on the tonnage band, different properties are evaluated and submitted to the European Chemical Agency (ECHA). Several toxicity studies need to be held, ensuring the safety of the production and the use of chemical substances. The requirements of the toxicity tests are very dependent on the tonnage band group that the company is producing and/or importing: (i) 1–10 t per year, (ii) 10–100 t per year, (iii) 100–1000 t per year, (iv) above 1000 t per year. The toxicity data obtained on each substance are shared within the group of producers/importers of that specific substance, under cost sharing principles. Exemptions of registration for research purposes are also possible via a PPORD (Product and Process Orientated Research and Development) application. When a company notifies ECHA with all the required information, the substance is then exempted from registration for the following 5 years, as long as the substance or article is only used for experimental trials and not used for commercial purposes, with no volume restriction.Polymer registration in EU: According to REACH polymers are exempted when the polymer producer has the registration or downstream use authorization for the monomers. This means that for the exemption of PEF, besides the registration of FDCA Avantium is also required to have a registration or a downstream user of MEG. Currently, the EU competent authorities are evaluating a regulation change to initiate the polymer registration processes and to define the group of polymers still exempted of registration when they are defined as polymer of low concern (PLC). Up to this day, REACH regulation has not been amended.

As mentioned, different jurisdictions require chemical registration according to different regulations: the UK follows UK-REACH regulation, which follows the principles of REACH since Brexit; in USA the registration of products is regulated by the Toxic Substances Control Act (TCSA) which supports EPA in the evaluation of the substances manufactured and imported in the USA; in Japan chemicals need to be registered in different governmental authorities such as the Industrial Safety and Health Law (ISHL) and Chemical Substance Control Law (CSCL). Before initiating the registration of the products, it is important to first evaluate the regions of interest either for production or for imports and evaluate the requirements of those registrations to prevent the need to repeat tests. Special attention needs to be given to the toxicological and ecotoxicological studies tests protocols that might be slightly different according to the requirements of the regulatory authorities.

FDCA is a substance registered by ECHA and Avantium is currently the lead registrant as the company initiated the registration, has the highest production volumes and is the owner of the toxicity studies. Currently the registration is up to a production volume of 1000 ton per year and will soon be updated to the tonnage band of higher than 1000 ton per year, well before the start-up of the Flagship Plant. FDCA is also listed in the USA and registered in Japan. The registration of FDCA and PEF in other regions of interest is a continuous activity at Avantium to ensure the globalization of the products.

The authorization of a material to be used in food contact application also requires a full assessment of the polymer to ensure the safety of the article to the consumers. In Europe, the Framework regulation EC No 1935/2004, in particular article 3, defines that any material or article intended to come into contact directly or indirectly with food must be produced in compliance to good manufacturing principles, shall be sufficiently inert to ensure that the their constituents are not transferred to food in quantities which could endanger human health or bring an unacceptable change in the composition of the food or a deterioration in its organoleptic properties. The good manufacturing principles are further described in the regulation (EC) No 2023/2006. The regulation (EC) No 10/2011 and its amendments, also known as the plastic regulation, establishes the specific requirements for the manufacture and marketing of plastic materials intended to be into contact with food. This regulation introduces the Union list, a list of authorized: (a) monomers or other starting substances, (b) additives excluding colorants, (c) polymer production aids excluding solvents, (d) macromolecules obtained from microbial fermentations. Non listed polymer production aids can be used in the manufacturing and are subject to national law. Non intentionally added substances and aids to polymerization (catalysts) may also be present in the plastic layer of plastic materials or articles. FDCA was added in 2015 to the Union list as a safe substance to be used in the manufacturing of PEF. Despite the substances being listed and/or approved by European or National laws, it is very important to ensure the starting substances are of the suitable purity for food contact applications. The understanding of potential impurities and the elimination/minimization of those impurities define the specifications of the starting substances and are key in defining the purification requirements of the technology. Pending on the application and intended use, the different players of the supply chain need to demonstrate the safety of their products by testing their product and evaluating the migrating species into food simulants. Food simulants are simplified matrixes that mimic the behaviour of the food (worse case scenarios) allowing the analytical evaluation of the substances that migrate form the plastic to the food. For example, if a company wishes to use the material in contact with clear drinks (e.g., water) a solution of ethanol 20 wt% in water is the food simulant and 3 wt% acetic acid in water testing needs to be added if the water is acidic (pH below 4.5). After the migration testing, the substances that migrate need to be investigated by different analytical techniques at very low detection limits. The known starting substances need to be investigated and need to meet the specification limits, but the non-intended substances also need to be identified and investigated if their migration levels meet safety limits, evaluating their chemical structure and deducing a safety limit by toxicity risk assessment, as defined by article 19. The non-intended added substances are starting substances, impurities and/or side products of the polymerization reaction (degradation products, incomplete polymerization substances) and they can be analysed by different analytical techniques to identify the high and low volatile substances. Usual techniques of LC-MS, GC-MS are used to identify and quantify the unknown substances. Based on the information gathered by the company, the compliance of the resin can be concluded and communicated to the supply chain via means of the declaration of compliance. Avantium has now developed a resin grade RP90Nx that is safe to be in direct contact with acetic foods and alcoholic drinks with an alcoholic strength less than 20% as well as with clear and cloudy drinks, incompliance with the European Regulations.

The European food contact regulations are currently being reviewed under the Farm to Fork strategy, as a result of the Green Deal implementation. Plastics regulation is focused on materials that are made from scratch. The European recycling regulations are also being updated and are expected to be implemented soon. As the regulation will bring a significant change of the recycling processes approval, no summary of the existing regulation is provided as it would be soon out of date.

The food contact approval system differs from jurisdiction to jurisdiction. While in Europe the focus is on the use of approved substances and evaluation of the safety of the constituents migrating to food, in USA Food Contact Notification focuses on the approval of a polymer linked to a manufacturing process and this approval is exclusive to the manufacturer. Beyond proving the safety of the polymer to be used as packaging of food, an environmental assessment, which is often related to recycling, needs to be provided to the food and drugs agency FDA. The Japanese regulation is currently moving from a voluntary basis organized by Japan Hygienic Olefin and Styrene Plastics Association (JHOSPA) to a Positive list principle organized by the Ministry of Health, Labour and Welfare (MHLW). The new regulation is expected to be fully in force by 2025. The regulation introduces four lists in two tables (content of tables not shown in this paper) of approved substances to be used as food contact materials:Table 1(1): Base Polymers (Plastics);Table 1(2): Base Polymers (Coatings);Table 1(3): Minor Monomers;Table 2: Additives.

PEF is listed as a synthetic polymer in the Base Polymer list of substances.

In conclusion, the approval of PEF as a food contact material in the different jurisdictions is a continuous effort of Avantium’s regulatory department to ensure the efficient globalization of the product.

### 1.6. Strategic Routes for Monetising Breakthrough Technologies

There are multiple strategic routes for monetising breakthrough technologies. These include
(i)Own and operate the technology;(ii)Applying the technology in partnerships or joint ventures;(iii)Licensing the technology to third parties;(iv)Or divesting the technology to third parties.

For each technology the preferred monetising routes are evaluated. Important criteria taken into consideration include among others: Operating and Capital expenditures of the plant, potential size of the market (e.g., licensing is only relevant when the potential market size is big), availability of suitable partners willing to engage under the right conditions, geographical location of the plant, IP protection, political climate, availability/need for soft money. Avantium has decided to build the 5000 tonne per year FDCA Flagship Plant as a majority shareholder. The production of PEF will be performed in existing assets with partners or via toll manufacturing. After building the Flagship Plant, Avantium has selected licensing to be the business model for the further roll-out of the FDCA and PEF technology: aside from being the most capital-efficient way to commercialise the technology, Avantium strongly believes it is also the fastest way to bring commercial quantities of PEF to market. With Avantium’s YXY^®^ Technology, brand owners and monomer- and resin producers and converters have the tools to significantly reduce their carbon footprint and to obtain access to a sustainable material with unique performance benefits. Avantium has identified two licensing scenarios for its potential clients:(i)Greenfield, where the licensee starts constructing the FDCA plant from scratch; and(ii)Retrofitted purified terephthalic acid (PTA) plants where the existing PTA plant is being converted into an FDCA plant.

For both scenario’s, Avantium will use its FDCA Flagship Plant to support the technology transfer.

For Ray Technology™, Avantium plans to form a joint venture with Cosun Beet Company (CBC), with the ambition to jointly construct and operate the first commercial plant for the production of plant-based glycols using Avantium’s Ray Technology™. The intent is that the joint venture will acquire a Ray Technology™ license from Avantium. As part of its licensing business model, Avantium will continue to develop and license its Ray Technology™ globally.

## 2. How the Technologies Developed over Time

### 2.1. Unlocking the Potential of a “Sleeping Giant”

Avantium was founded in 2000, with the objective to accelerate and exploit the application of high-throughput catalysis research. However, the discovery of Avantium’s lead product FDCA, the building block for PEF, goes back to a Friday afternoon in 2005. One year earlier, Avantium made the strategic decision to leverage its expertise in high-throughput catalysis R&D by initiating its own proprietary development programmes focused on biobased chemicals, materials and fuels in light of the upcoming transition into renewable feedstocks and sustainable materials. After several brainstorm sessions, 5-(hydroxymethyl)furfural (HMF) was identified as a very interesting fuel precursor, but the synthesis and isolation of this molecule had proven to be a serious problem and a different approach was required. To this end, Chief Technology Officer Gert-Jan Gruter and Scientist Erik-Jan Ras performed a small experiment on a Friday afternoon in 2005 with some sugar (sucrose) from the company restaurant. Instead of using water as a solvent (water was the typical solvent of choice during the previous 2 centuries of sugar chemistry), they mixed the sugar with alcohol (ethanol) and some acid in order to dehydrate the sugar at 150–200 °C. The result of this effort was not HMF but a beautiful furanic molecule: 5-(ethoxymethyl)furfural (EMF). Gert-Jan Gruter immediately realised that this modified dehydration could be a breakthrough invention, due to the large potential of this intermediate towards fuels and FDCA and FDCA-based polyesters such as PEF.

FDCA was listed already in 2004 by the US Department of Energy as the #2 in the top-12 priority chemicals for establishing the “green” chemistry industry of the future and has remained this prominent position over the years [20,21,30,31]. Given the huge potential of FDCA, industrial production of this monomer building block was pursued and researched for over 100 years in the laboratory, however without successful scale-up. As such, FDCA has been called a “sleeping giant”; “sleeping” because no one had ever succeeded in producing FDCA in an economic fashion, and “giant” because of its enormous market potential. Gert-Jan Gruter figured out why no one was able to make this chemistry work: everyone used water as a solvent for the first sugar dehydration step. He came up with the simple but revolutionary idea to run the process in alcohol. This “Eureka! moment” was the beginning of Avantium’s most advanced technology, the plant-to-plastics YXY^®^ Technology.

The YXY^®^ Technology is a new and innovative way to produce FDCA. It comprises a chemical catalytic process to produce RMF (5-(methoxymethyl) furfural with a certain amount of HMF), which is subsequently converted into FDCA in an oxidation step. Avantium is the first company to develop an economically viable route to producing FDCA at large industrial scale, an important future monomer building block. This was achieved by applying Avantium’s expertise of catalyst- and catalytic process development using a high throughput testing platform [1]. Deshan and co-workers recently reviewed the different routes towards FDCA [32]. The technology to produce PEF can be divided into several catalytic steps, of which the following are the most important (see Figure 4):Step 1:Sugar dehydration. The catalytic dehydration (i.e., the removal of oxygen via water elimination) of plant-based sugars (high fructose syrup) in an alcohol, to make an alkoxymethyl furfural such as methoxymethyl furfural (MMF). Van Putten et al. has extensively reported about the chemistry involved in the conversion of carbohydrates into furanic compounds [3,33,34,35,36,37];Step 2:Oxidation the catalytic oxidation of an alkoxymethyl furfural (such as MMF) in acetic acid to make furan dicarboxylic acid (‘crude’ (c)FDCA). The similarities and differences of the conversions of para-xylene into terephthalic acid compared with the oxidation of RMF into FDCA have been extensively discussed by van der Waal et al. [38,39,40,41,42];Step 3:Purification, removal of product impurities via purification producing purified FDCA (pFDCA) [43,44,45,46,47];Step 4:Melt polymerization of FDCA and mono ethylene glycol (biobased MEG) to create the plant-based polymer, polyethylene furanoate (PEF) [5,22,48,49,50,51,52]. Typically, the melt polymerization is followed by a solid-state polymerization step to bring the polymer molecular weight to the desired values, depending on the target application(s) [53,54];Step 5:Processing of PEF into bottles (injection stretch blow moulding (ISBM) [22], trays (extrusion and thermoforming) [29], fibres (melt spinning) [26,55,56,57,58,59], and films (extrusion, tenter stretching and lamination) [22,23,24,25,27,28,29,46,60].Step 6:Mechanical or Chemical recycling of PEF [61,62];

**Figure 4 polymers-14-00943-f004:**
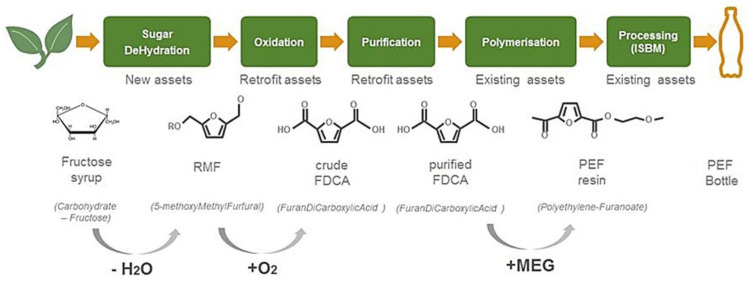
Conversion steps to transform fructose into FDCA and subsequently PEF.

Steps 1, 2 and 3 have been proven in the pilot plant and will be incorporated in the Flagship Plant. The polymerization to PEF has been successfully proven at commercial batch scale in Japan and in Europe and will be performed in existing assets with Avantium partners, depending on the location of the customer.

FDCA can be applied into many other products in addition to be a building block for PEF. In principle, all applications now using terephthalic acid or one of the other diacid isomers can be replaced by FDCA. Not always the technical benefits which are encountered by replacing PTA by FDCA to make PEF instead of PET are encountered but when FDCA becomes an economically competitive bulk monomer those other outlets will also be very interesting. Many other FDCA containing polyesters have been reported as well as polyamides, polyurethanes, thermoset resins [7,8,63,64,65,66,67,68,69] and references therein] as well as plasticizers to replace phthalate-based compounds [70,71,72].

### 2.2. Sugar Dehydration into MMF/HMF and Oxidation into FDCA

The use of carbohydrates as building block for chemicals and fuels poses an interesting challenge. Since sugars are very oxygen rich and rather low in energy content and petro-based chemicals are almost devoid of oxygen, it is clear that oxygen removing reactions are often needed. With sugars, this can be performed through three major pathways: decarboxylation/decarbonylation, hydrogenolysis using hydrogen or dehydration. No matter what route one chooses, the loss of mass will be considerable, and this always has cost implications. Van der Waal and de Jong [35] showed that many emerging carbohydrate-based technologies inherently require considerable amounts of carbohydrates, often multiple kg per kg of product even in the ideal chemistry case, and thus can only be economically competitive for drop-in molecules when cheap carbohydrate sources can be used. Examples for atom inefficient conversions are sugars to ethylene or p-xylene, both requiring (even in the best case) more than 3 kg carbohydrate to produce 1 kg of product.

At Avantium the acid-catalysed dehydration route of C6 sugars was chosen to be the focus of the research. Fructose and to a lesser extend glucose are converted at elevated temperatures into hydroxymethylfurfural, or HMF, which was long known but despite two pilot plant projects never commercialized [33].

The development of a commercial process for HMF would certainly allow a technological breakthrough although HMF in itself is not a tremendously convenient building block. It has two polar groups, an alcohol group and an aldehyde group, and thus does not dissolve readily in the typically apolar solvents. In addition, HMF is rather unstable and therefore difficult to purify, reacting further to black insoluble compounds (humins) or it reacts with water to levulinic acid.

It was here that Gruter and Dautzenberg realized that much better building blocks would be obtained if the alcohol and/or the aldehyde group would be converted directly to much less polar groups such as ethers, esters and acetals. Moreover, the conditions that would be required for etherification and esterification would be very similar to those of the dehydration, i.e., acidic catalysts and elevated temperatures. Thus, by replacing water as the solvent in the dehydration reaction either by an alcohol or by a carboxylic acid, one would convert the hydroxy group into an ether or ester, respectively. With the principal conversion of fructose to MMF in methanol was shown to work, it was time to develop an economically viable route, and at this stage a Conceptual Process Design (CPD) was performed. In a CPD, an initial design of a commercial-sized plant is modelled on the basis of the best catalytic results and process conditions obtained. Assumptions for the best reactor configuration and a work-up section were obtained by READPERT and SIMPRO packages. The conceptual plant is then modelled in ASPEN and appropriate heat integration between units is applied. At this point in time, this model is very useful for directing the subsequent research needed, as sensitivity analysis on several process parameters can easily be performed. This method quickly identifies the most important process variables and sets clear targets for the research to be performed. The CPD analysis for the MMF process clearly showed that for an economically viable process it was required to increase the fructose concentration in the feed while attaining or even improving the yield of the main products. Though these targets do not seem too surprising, the CPD now allowed Avantium to quantify these targets with tangible numbers. Though seemingly logical to increase the fructose concentration in the feed on paper, this proved to be less easy as the solubility of sugars in organic solvents is rather limited [73,74]. Sugars, fructose not being an exception, generally tend to dissolve only well in very polar solvents like water, DMSO, DMA and the like. Solubility of fructose and other sugars in alcohols have been reported for methanol and other alcohols [73] (and references therein). Clearly, methanol was the solvent of choice, but even then, solubility of fructose was still too low based on our conceptual process design. Solubility in methanol can be improved by increasing the amount of water in the solvent [73], though high amounts of water are catalytically undesired as it hinders the etherification of HMF to MMF and enhances the rehydration of the furanic intermediates to levulinic acid. At the same time, the CPD showed that it was economically undesired to remove all water (formed from the fructose dehydration) from the methanol recycle. However, a real break-through came when it was realized that it was not necessary to have fructose in its sugar form in the reaction [37]. Over the course of the catalytic experiments, it was often noted that the fructose was in equilibrium with its methyl fructosides. These methyl fructosides can be synthesized readily at low temperatures in methanol, and to our surprise are a fully miscible liquid with methanol in high concentration. Using the fructosides allowed us to obtain the desired high concentration of sugars in methanol [37]. To better understand the reactivity of sugars towards conversion into HMF and MMF, four different ketohexoses (fructose, tagatose, sorbose and psicose (only in methanol)) were converted into 5-hydroxymethylfurfural (HMF) was explored in water [3] as well as methanol [4] using sulphuric acid as the catalyst. Significant differences in reactivity were observed both in water as well as in methanol. In methanol, tagatose and psicose were clearly more reactive than fructose and sorbose. However, the selectivity to MMF was found to be higher for fructose and psicose than for tagatose and sorbose while 2-Hydroxyacetylfuran (HAF) and its corresponding methylether was shown to be a by-product for mainly sorbose and tagatose (as high as 8% yield). The results indicate that the relative orientation of the hexose hydroxyl groups on C3 and C4 has a major effect on both the reactivity and selectivity. This suggests that the dehydration towards HMF takes place via a mechanism with cyclic intermediates in which the C3-C4 bond is fixed in a ring structure. The reactivity of the sugars was significantly lower in water [3] than that observed in methanol [4]. So, the best carbohydrate source for the production of MMF is psicose, unfortunately it is a very rare sugar and at the short term not available at big quantities.

Now having developed a process for the production of the MMF intermediate, the oxidation of MMF was investigated. Though several methods have been reported for HMF oxidation, the presence of the methoxy group in the MMF molecule offered an interesting challenge as it is not easily oxidized.

Many oxidation catalysts and processes were evaluated using our high-throughput parallel batch screening equipment, but in the end the only system that gave good yields in FDCA was the Co/Mn/Br in acetic acid system [39,40]. Interestingly, this catalyst system is a well-known industrially applied catalytic oxidation system for the oxidation of p-xylene to terephthalic acid, the molecule FDCA potentially seeks to replace. Although FDCA is not a drop-in replacement for terephthalic acid, one could consider the existing industrial oxidation plants as drop-in technology assets for future FDCA plants.

### 2.3. Humins and Methyl Levulinate, Side-Products in the MMF Process

In Step 1 of the conversion process of fructose into FDCA, besides MMF also co-products are formed (Figure 4). Noteworthy are the production of humins and methyl levulinate. Humins are heterogeneous condensation by-products produced by random polymerization processes during the acid-catalysed dehydration of fructose as well as glucose (the feedstock is typically a fructose/glucose 90/10 or 95/5 mixture). The formation of humins is a fundamental drawback of the water- or alcohol-based acid catalysed dehydration process, resulting in reduction in yield of the primary product (MMF/HMF), reactor fouling, and other engineering challenges [75,76]. Humins are almost unavoidably formed during acid catalysed conversion of the carbohydrate fractions of biomass to produce furan-based chemicals. Rehydration of MMF/HMF under acid conditions leads to the formation of methyl levulinate/levulinic acid and methyl formate/formic acid.

#### 2.3.1. Humins

From a technical as well as from an economic viewpoint, four strategies are possible to cope with the challenges linked to the production of humins:(a)Optimizing the acid catalysed dehydration process targeting the minimization of humins production;(b)Adapting the acid catalysed dehydration process targeting the composition of the humins production. The use of an alcoholic solvent in the MMF process results in a highly viscous liquid humins (after solvent evaporation), instead of solid humins encountered in water-based dehydration systems;(c)As a base case, the conversion of humins into a heat and power source to satisfy a substantial part of the energy demand of the biorefinery is pursued;(d)A more favourable longer-term strategy is using the humins as a potential valuable, renewable feedstock for new biobased chemicals, biomaterials, and/or additives of interest.

This overview will summarize several of the most potential valorisation routes of humins. In the most recent few years, academia and industry have showed an increasing interest for developing applications involving humins. Different approaches have been suggested, including the use for carbon-based materials, as cross-linking agent to obtain valuable functional materials as well as to depolymerise the humins to obtain valuable chemicals. Humins are carbon-rich polymers containing about 50–60 weight % of carbon in their structure and can therefore be regarded as a potential feedstock for producing activated carbon; extensively used in wastewater treatment applications. Several investigations have reported the use of humins as a substrate for impregnation material. Humins can be non-reversibly converted upon heating into structures with an increased glass transition temperature (Tg > 65 °C), making them good materials for thermoset-like resins [77,78,79] and into nanocomposites [80]. Pin et al. (2014) proposed the development of biobased polymeric materials containing humins as a component into a polyfurfuryl alcohol (PFA) network to generate homogeneous materials and compared them with PFA/lignin, the humins containing matrix resulted in higher tensile strength [81,82]. Humins can also be used for wood modification [83]. The burning behaviour and thermal hazards of humins have been established [84,85] as well as the ecotoxicological aspects of the use of humins in the environment [86]. The behaviour of humins after different thermal treatments have been evaluated in detail and shown to be tuneable [87,88]. Tosi et al. [89,90,91] showed that under the right conditions the auto cross-linking behaviour of humins can lead to thermoset porous carbon materials called humins foams with tuneable properties. One common feature of humins, as a polyfuranic thermoset material, is their inherent brittleness which is a direct consequence of the network’s structure. Auto-crosslinked humins networks exhibit only minor deformation and break very easily. Consequently, this behaviour limits their use in many industrial applications. The group of Mija has overcome this limitation by combining humins with epoxide based aliphatic ethers, as a toughening approach, resulting in humins copolymers that have a ductile and elastomeric character [92,93,94]. A recent publication discusses the use of humins as a biobased binder in asphalt as a partial replacement of bitumen [95,96].

#### 2.3.2. Methyl Levulinate

Another side product in the MMF process is methyl levulinate (ML), the methyl ester of levulinic acid (LA) [97,98]. Methyl levulinate can be used as a solvent, in fragrances, as a plasticiser or as a natural herbicide [97] but can also function as a building block for numerous biobased applications in, e.g., cosmetics, food preservatives and fuels [99]. The importance of ML/LA as building block was first highlighted by Werpy and Pedersen [20] and later revised by Bozell [30]. Currently, several companies are either offering LA or build on levulinates as chemical building blocks to create a wide array of products [21]. Recently other outlets have been reported including the conversion into the solvent 2,2,5,5-tetramethyloxolane (TMO) [100] as well as the catalytic conversion into methyl acrylate/acetic acid [101] or into methyl vinyl ketone (MVK) [102]. An overview of the potential products that can be obtained from levulinic acid is given in Figure 5.

### 2.4. Flywheel for Commercial Developments

Since the discovery of this novel MMF production route in 2005, Avantium has evolved into a world-leader in FDCA and PEF. Avantium believes it was the first in 2009 to test PEF in a wide range of applications, such as bottles, fibres and films. In 2011, Avantium was the first company to start construction and operation of an MMF and FDCA pilot plant, operational 24/7 (Figure 6). The objective of a pilot plant is to scale-up the technology from lab to a scalable demonstration size, to further optimise the technology and produce product to validate applications, serving as a flywheel for commercial developments. The FDCA pilot plant has enabled Avantium to produce many tonnes of FDCA and PEF samples that are representative of the final product from subsequent commercial plants. Furthermore, the pilot plant enabled Avantium to test PEF in various applications both in-house and through its partners. In 2015, FDCA was approved by the European Food Safety Authority (EFSA). In August 2016, FDCA was included in the Plastics Regulation as a food contact material. In 2021, Avantium released PEF food contact grade, RP90Nx, that complies with the required regulations of food contact materials, which would allow the use of the PEF resin in direct contact with acetic foods and alcoholic drinks with an alcoholic strength less than 20% as well as with clear and cloudy drinks in the European Union and the UK. Moreover, Avantium requested the Technical Committee of the European PET Bottle Platform (EPBP) to conduct an evaluation of the effect of PEF on the PET recycling stream. Based on this assessment, EPBP has awarded interim and conditional endorsement to Avantium’s PEF polyester resin in a test market (up to 50 kt/a), with a limitation of a maximum market penetration of 2% in 2017 [103].

### 2.5. Revised Scale-Up and Market Launch Strategy

In 2016, Avantium established the 49:51 joint venture Synvina with BASF to commercialise the YXY^®^ Technology. Apart from technological progress that led to technology completion, the Synvina joint venture did not result in the successful collaboration that was envisioned at its start. Due to differing views on the commercialization strategy of FDCA and PEF, BASF and Avantium decided to dissolve the Synvina joint venture in January 2019. When dissolving Synvina, Avantium bought back the joint venture shares from BASF and acquired 100% ownership of the YXY^®^ Technology again. Thereafter, Avantium explored different scenarios with potential partners and customers to redefine the commercialization strategy of PEF to meet both market and capital requirements. This led to a revised scale-up and market launch strategy presented by Avantium in June 2019. Avantium announced its intent to construct and operate a 5000 tonnes per annum FDCA Flagship Plant. The reduction in the production scale required a re-evaluation of the process steps and some redesign of the technology, as well as a better integration with the business needs. This becomes now a reality: on 9 December 2021 Avantium announced its final investment decision (FID) with which it is now on the brink of commercialising its most mature technology, with the goal to unlock the large potential of the FDCA and PEF [104].

## 3. How It Is Going: PEF Has the Potential to Revolutionize the Plastic Packaging Industry

Through the many years of extensive research and development and rigorous testing in the pilot phase, Avantium has demonstrated PEF to offer superior packaging solutions in performance and environmental benefits when compared to conventional fossil-based commodity plastics. PEF offers a unique solution to address the global need to reduce plastic waste, help tackle climate change and transition into a circular, sustainable biobased economy.

### 3.1. The Need to Keep Fossil Resources in the Ground—And Only Use Carbon Sourced above the Ground

The climate crisis and plastic waste pollution provide a sobering but relevant background for Avantium’s work and strategy to bring PEF to market. Climate change is one of the most pressing issues of our generation. In 2015, 196 heads of state and climate experts agreed under the United Nations Paris Agreement to limit global warming to 1.5 °C, compared to pre-industrial levels. Beyond a 1.5 °C change there will be so much heat globally to push many of the planet’s natural systems out of balance; a balance that cannot be regained. This was recently reaffirmed under the 2021 Glasgow Climate Pact.

The 2021 report from the Intergovernmental Panel on Climate Change (IPCC) revealed that CO_2_ emissions were still higher than at any time during the last two million years. It is therefore unequivocal that human-caused emissions are a significant factor in climate change [105]. The global climate breakdown demands an entirely new way of doing business, moving the world from its dependence on fossil-based resources towards a sustainable future with renewable energy and -materials at its basis.

In recent years, public concern has grown around the damaging levels of CO_2_ emissions from plastic production and the large quantities of plastic waste polluting our oceans. In 2019 the carbon footprint of plastics was 860 million tonnes (0.86 Gt). With the growing plastic demand (3.5% average volume growth per year), the carbon footprint will grow to 1.34 Gt in 2030 and 2.8 Gt in 2050 if we continue to use fossil feedstock for our plastics [106] and if we do not improve the recyclability of our largest volume plastics (Figure 7).

At the same time, it was agreed to reduce the global fossil-based CO_2_ emissions from 36.8 Gt in 2018 with 90% to 3.7 Gt in 2050. It is clear from the above numbers that without a plastics materials transition in which we transition to use carbon from above the ground (biomass, CO_2_ and recycle) for producing our plastics, we will not make the 2050 CO_2_ emission targets.

Most plastics have a useful but very short life and create significant and long-lasting harm for our planet, both during manufacture and after disposal. As a society, we therefore need to radically and urgently transform the way we produce, use and discard plastics. In essence, society has to move from today’s linear model to a circular use of plastics. The chemical and plastics industry need to employ more sustainable practices and create products and materials that feed into the circular economy. This will in turn reduce plastic pollution and overall carbon footprint. The world needs to embrace a business where we use renewable sources of carbon, emitting no fossil carbon into the atmosphere during production and disposal.

There is a general global consensus on the need to move to a 100% renewable—and decarbonized—energy sector, for example by using solar, wind or hydropower sources. However, there is not yet an equivalent strategy for the materials sector, where carbon is essential. This is especially true for the chemical and plastic industries, where progress away from fossil carbon sources towards above-ground carbon sources has been slow. Under a renewable carbon strategy for the chemical industry, manufacturers would need to stop using fossil (geosphere) sources and instead use the renewable carbon, such as glucose or sucrose from plants or the carbon found in CO_2_. Therefore, industry has to go beyond using renewable energy. All fossil carbon use has to end, as the carbon contained in the molecules of chemicals and plastics is prone to end up in the atmosphere sooner or later. Only a full phase-out of fossil carbon will help to prevent a further increase in CO_2_ concentrations. The Renewable Carbon Initiative is a recent initiative to guide this transition bringing together stakeholders all along the value chain [107]. Solutions such as PEF will be crucial for the chemical and plastics industry to move away from fossil-based resources and embrace sustainable, circular technologies to enable a circular, greener future.

The 55% recycling indicated in the NOVA scenario of Figure 8 would require a shift from predominantly polyolefins as commodity plastics (PE, PP, PS, PVC) to predominantly polyesters. Next to the fact that polyesters have much better atom efficiencies when produced from sugars (more than 50% oxygen content) and CO_2_ (75% oxygen content), polyesters are the only plastics that can be closed-loop recycled (both mechanical- and chemical recycling) with very high yields (e.g., bottle-to-bottle PET mechanical recycling); see below.

### 3.2. PEF Helps Tackle Climate Change and Addresses the Global Need to Reduce Plastic Waste

The 100% plant-based, fully recyclable PEF is designed to improve the circularity of plastics. PEF is 100% made from the sugars generated by plants. The sugars (fructose syrup) used today to make FDCA can be produced from agricultural crops, such as wheat, corn and sugar beet. When commercially available, PEF can also be produced from cellulosic sugars, which are abundant in non-edible biomass, such as agricultural and forestry residue streams.

#### 3.2.1. PEF in the Circular Economy

PEF, being a polyester has some distinct benefits with respect to circularity when compared to other polymers. It has proven fit-for-purpose with existing sorting and recycling facilities. PEF can be recycled mechanically and chemically using the same technologies that are existing or under development for PET recycling (Figure 9). In addition, PEF can easily be distinguished and sorted from PET and other plastics using a commonly employed NIR sorting technique, which is a prerequisite for enabling the initiation of a closed PEF recycling loop, either in the form of a packaging (re-use), of a polymer (mechanical recycling) or monomer/prepolymer (chemical recycling).

#### 3.2.2. Re-Use

The implementation of re-use plastic packaging solutions (mainly refillable bottles) has been scaled down in the past decades, but the interest to develop re-use plastic packaging is now reviving as a consequence of the drive towards circularity. Avantium is also involved in scoping and developing FDCA based re-use packaging solutions.

#### 3.2.3. Mechanical Recycling: Closed Loop

The most favourable form of recycling is closed loop mechanical recycling as it keeps the material in the loop at the same quality level. This requires a method to revamp the molecules after each loop to counteract the losses in quality the polymer encounters each loop, most notably due to loss in molecular weight upon melt processing. Alike PET, PEF is a polyester that has end groups at its polymeric chain that can still react. This gives the possibility to regain the molecular weight upon a so-called Solid-State Polymerization (SPP) process with a polycondensation reaction. This SSP step therewith can compensate for typical molecular weight losses polymers encounter in melt processing steps required to make final articles. Such regain in molecular weight is not possible for the largest polymer family (in market volume): polyolefins. Polyesters like PEF and PET therefore have the inherent advantage that the molecular weight of the recycled polymer can be kept at a constant level, loop after loop; a crucial property to allow a material loop without the need to go back to monomers or polymer precursors. With a constant molecular weight, it is possible to keep the processing and performance behaviour of the recycled resin close to that of the virgin resin. In practice however, due to losses in the loop and the need to prevent accumulation of contaminants in the material loop, there will always be a need for a (minor) inflow of virgin resin or, in the future, a combination with a chemical recycling step for a reduction in accumulated impurities. The similarity between PEF and PET is such that PEF can be recycled in existing PET mechanical recycling assets such as, dryers, extruders, crystallizers and SSP equipment.

#### 3.2.4. Mechanical Recycling: Open Loop

Open loop recycling can mean either that PEF is lost in another recycling stream such as that of PET or that the type of end product made form the recycled PEF is different than the end product the recyclate originated from. A potential cause for the first is the use of PEF as a barrier material either in the form of a blend or multilayer in PET and for the latter the most encountered route in PET is the use of recycled PET in the fibre industry, in particular that of textiles.

Obviously, the blending of polymers during recycling should be avoided as much as possible from the circularity perspective. However, for some applications use of barrier material alongside PET is required to meet the high standard of barrier performance. As an example, many multilayer PET bottles currently include semi-aromatic polyamide (PA) resins for improved gas barrier properties required for small volume beer, wine and carbonated soft drinks PET bottles. However, a common issue for incumbent barrier materials such as PA and EVOH is the detrimental effect they can have on the processing and performance of the resulting rPET when these polymers enter the PET recycling loop even at low concentrations. In case PEF does enter the PET recycling process, for example, due to sorting errors, PEF has the beneficial property of being able to transesterify with PET as has also been reported by Papageorgiou [108]. This means the FDCA will be built in the PET chains much alike commonly used comonomers in PET grades like purified isophthalic acid (PIA) or cyclohexanedimethanol (CHDM). This results in haze free co-polyesters even at high PEF levels [109,110,111]. The influence on rPET when PEF would enter the PET recycling stream has been assessed according to EPBP protocol for monomaterial bottles (resulting in the previously mentioned interim endorsement). In the other type of open loop recycling where PEF would enter a different application field Avantium has demonstrated to be able to spin fibres and produce a t-shirt from rPEF.

#### 3.2.5. Chemical Recycling

Some waste streams will be too contaminated or looped so many times that a more rigorous recycling step is required. In those cases, a depolymerization, filtration/purification and repolymerization step, also referred to as chemical recycling, or tertiary recycling, can result in the desired quality of the resulting polymer. The term chemical recycling is used for a wide variety of processes that have polymers as an input and an oligomer, monomer or even precursors of monomers as an output. In the case of polyesters, the most studied routes are several solvolysis routes, where the polyester is broken down in reaction with a solvent, typically at elevated temperatures. The most common routes of solvolysis of polyesters are glycolysis, methanolysis and hydrolysis (acidic, neutral or alkaline environment). So far only limited research on chemical recycling of PEF has been published [112].
The glycolysis route is the most implemented route for PET. In this process the addition of a glycol (typically MEG) and a transesterification catalyst at elevated temperatures converts PET into oligomers (also referred to as pre-polymers) such as bis-hydroxyethyl terephthalate (BHET). These pre-polymers can be fed in the melt polymerization process going back to PET (closed loop 3° recycle) but can also be applied as feedstock for other polymers like polyurethanes, thermoset resins, etc. (open loop 3° recycle). Little research has been published on the glycolysis of PEF, although Gabirondo et al. have demonstrated it is possible to apply a glycolysis process to PEF [113].The methanolysis route is considered to allow the best purification of the end products from contaminants. The addition of methanol leads to formation of dimethyl terephthalate (DMT) and MEG when starting with PET. Sipos et al. demonstrated that the methanolysis of PEF into dimethyl furanoate (DMF) and MEG can reach higher conversion rates than that of PET [40].In the hydrolysis route the polyester is broken down by reacting with water. Depending on the acidity this process can be sped up and carried out at milder conditions. The advantage of the hydrolysis route for PET is that it goes back to terephthalic acid (TA), with the downside that the purification of the TA is challenging. Sipos et al. have reported a yield >80% for a first acidic hydrolysis assessment on PEF, demonstrating that the principle can be applied to recover FDCA from PEF [61]. The alkaline hydrolysis of PET/PEF co-polyesters has been described by Vinnakota [114].Besides the chemo-catalytic solvent routes various papers have reported on the enzymatic depolymerization of PEF [115,116,117] and reviewed by Loos et al. [8].

At the current status of knowledge there is no indication that the chemical recycling methods under development for PET are not equally adaptable for the chemical recycling of PEF.

#### 3.2.6. End-of-Life

Another feature of PEF is that it degrades much faster than conventional plastics when exposed to fungi and bacteria under industrial composting conditions, as has been assessed by the Belgian company Organic Waste Systems [55]. The testing under industrial composting conditions was **NOT** done to evaluate the suitability of PEF for industrial composting as a designed end of life. These tests were performed as we wanted to learn if any degradation would occur with PEF (in comparison to the known undegradable PET) and industrial composting conditions (58 °C) were selected as these experiments would be much faster than ambient temperature degradation experiments. In 2018, also field trials with oriented and amorphous PEF film on the Avantium balcony in Amsterdam were started. Enough parallel experiments were started to allow for a 6-monthly sampling in triplicate, involving the analysis of sample weight loss, changes in molecular weight, electron microscopy, thermal analysis, etc. First results are expected in the near future.

Figure 10 shows clearly that PEF is converted much faster into CO_2_ than PET, which was considered non degradable under these conditions. After 60 days 90% of the cellulose reference was converted into CO_2_; weathered and unweathered PEF took 240 and 385 days, respectively, to reach 90% biodegradation. Furthermore, initial studies suggest that PEF degrades in the natural environment many times faster than PET—it starts to break down in natural conditions within a single year instead of hundreds of years. Slow degradation in the environment is important as it will avoid the endless accumulation of waste plastic over many decades or even centuries which is the case with non-degradable materials such as PET.

### 3.3. Superior Functionality

PEF and PET are two polyesters with close chemical structures. However, the thermal, mechanical and barrier properties are different (Table 1, Figure 11). The many trials with partners over the past years have also resulted in much better insights in the performance of PEF. Avantium and its partners found out that PEF has high performing gas barrier properties for carbon dioxide (CO_2_) and oxygen (O_2_) compared to conventional plastics, leading to a longer shelf life of packaged products (Table 1 and references therein). This makes it possible to fully enjoy products such as food, drinks and cosmetics -even if stored for a long time—avoiding unnecessary food and product waste. This is an important feature, as food waste is a widely known global problem and a reduction can result in significant reduction Green House Gas (GHG) emissions [118,119,120]. Plastics—if used, produced and discarded responsibly—can help combat this problem. PEF offers even better barrier properties than conventional plastics and is therefore a highly sustainable option for shelf-life extension.

The reason why PEF has such a good barrier is quite interesting. The overall gas permeability is a function of both sorption and diffusion. Interestingly, the PEF matrix has a higher CO_2_ sorption than PET due to the polar moment of the furan ring which could favourably interact with polar molecules [121]. However, the permeability is strongly reduced (Table 1). This is explained by the very limited local motions in PEF, such as the hindered furan ring flipping together with the restricted carbonyl rotations that decrease diffusion of small molecules [121,122,123,124,125,126,127]. It is important to highlight that the lower permeability of PEF compared to PET is also preserved in biaxially-stretched samples, as also the permeability of PEF goes down with an increase in (strain induced) crystallinity [25,128].

PEF also offers higher mechanical stiffness and strength. In combination with the higher barrier properties, this higher mechanical strength can allow downgauging for certain packaging solutions where barrier and mechanics are crucial, leading to weight reductions up to and over 20%, in line with the ambitions on packaging waste reduction by the EU.

In terms of thermal properties, solid, amorphous PEF has an improved ability to withstand heat (~10 °C higher glass transition temperature than PET) and can be processed at lower temperatures (~30 °C lower melting point than PET) (Table 1). PEF has a higher stiffness and yield strength when compared to those of PET, which allows for increased shaping possibilities. PEF’s chain and crystal structure [127,129,130,131,132] as well as its quiescent crystallization kinetics [133,134,135,136] has been researched extensively. For the performance of end products another type of crystallization is of interest: strain induced crystallinity (SIC). Some polymers can form crystals upon orientation of the polymeric chains above its glass transition and many shaping processes make use of this principle: bottle blowing, biaxial film stretching and fibre drawing. The SIC formation behaviour of PEF has received attention in the last decade in the scientific literature. Not only the crystalline structure of SIC [129,130,131,137], but also the formation of SIC and the influence of stretching conditions, differ from that of PET [132,138,139,140]. This difference is mainly attributed to the lower entanglement density of PEF [128], meaning that PEF needs to be stretched more before a similar amount of orientation is built up. Moreover, due to the more ordered crystalline form (the crystal unit cell of PEF has two repeating unit as opposed to one in PET), the orientation required to form a crystal is expected to be larger. As a consequence, PEF requires higher draw ratios before a similar amount of SIC is formed. Although this does have consequences on for example the optimal preform geometry when forming a specific bottle, or the optimal conditions under which the blowing takes place, the processing of PEF can be performed on the same equipment as employed for PET moulding and stretching. The important consequence is that even tough PEF is not a drop-in for PET, a PET converter does not need to invest in new equipment when adding PEF articles to its product portfolio.

**Table 1 polymers-14-00943-t001:** Comparison of some of the major characteristics of the polyesters PET and PEF.

Property	PET (Amorphous)	PEF (Amorphous)	References
Molecule	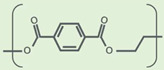	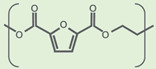	
Density (amorphous)	1.36 g/cm^3^	1.434 g/cm^3^	[129,130,133,141]
Density (crystalline, calculated)	1.455 g/cm^3^	1.565 g/cm^3^	[129,130,142,143,144]
Melting temperature (T_m_)	250–270 °C	210–230 °C	[56,109]
Glass transition temperature (T_g_)	~76 °C	~88 °C	[145,146]
Crystallization time	2–3 min	20–30 min	[128,133,134,135,147]
E-modulus (ISO 527/1A, 1 mm/min)	2.1–2.2 GPa	3.6 GPa	[146]
Yield strength (ISO 527/1A, 10 mm/min)	50–60 MPa	90–100 MPa	[146]
O_2_ permeability * (@23 °C, 65% RH)	2.5 cm^3^·mm/(m^2^∙24 h∙bar)	0.23 cm^3^·mm/(m^2^∙24 h∙bar)	[146]
CO_2_ permeability * (@23 °C, 0% RH)	23.6 cm^3^·mm/(m^2^∙24 h∙bar)	1.6 cm^3^·mm/(m^2^∙24 h∙bar)	[146]
H_2_O permeability * (@38 °C, 90% RH)	0.9 g∙mm/(m^2^∙24 h)	0.36 g∙mm/(m^2^∙24 h)	[146]

* All permeability experiments were carried out by an independent laboratory on 45 µm thick cast films in accordance with ASTM: ASTM F1927-14 (O_2_), ASTM F2476-13 (CO_2_) and ASTM F1249-13 (H_2_O). For PET a commercial bottle grade PET was used with an IV of 0.80 dL/g and the PEF resin used had an IV of 0.89 dL/g as measured according to ASTM D4603 (0.4 g/L).

### 3.4. Sustainability

A decade ago, Eerhart et al. showed the high potential of PEF by an initial process analysis, energy and GHG balances for the production of FDCA and PEF from 1st and 2nd generation feedstocks [6,9,11]. A review of the different LCA’s performed for the production of HMF, FDCA and PEF was recently pursued by Davidson et al. [148]. Avantium partnered with nova-Institut GmbH under the framework of the PEFerence project, to perform a full cradle-to-grave Life Cycle Assessment (LCA) for a 1st of a kind FDCA plant based on the YXY^®^ Technology, assessing the potential environmental impacts of a 250 mL PEF bottle packaging solution in comparison to conventional PET alternatives. This LCA has been conducted compliant with the standards ISO 14040 and 14044. A critical peer review of the study, including experts on LCA methodology as well as on incumbent packaging solutions, was conducted in order to verify whether the LCA met the requirements for methodology, data, interpretation and reporting. In Figure 12, a comparative analysis of the main Life Cycle Assessment indicators for 250 mL PEF and PET bottles is presented.

The following conclusions are highlighted from the comparative analysis:The use of 100% renewable carbon in PEF instead of fossil carbon in PET for producing 250 mL bottles would result in significant reductions in greenhouse gas emissions (−33%) over the life cycle of the bottles.PEF bottles would also contribute to remarkably less finite resource consumption of fossil fuels (−45%) compared to that demanded by PET bottles.These impact potentials are two of the most relevant environmental impact categories in the current political agenda driving the transition from fossil to renewable carbon. This represents a significant benefit, because climate change and resource use were found to be the impact categories most heavily influencing the environmental impact of monolayer PEF bottles.Very significant is the lower pressure that the production of PEF bottles would put on abiotic resources (minerals and metals) in contrast to that caused during PET bottles production.The lower environmental footprint of the biobased alternative can be attributed, to a great extent, to the improved barrier and mechanical properties of PEF allowing for an overall 46% reduction in polymer usage in the manufacture of bottles. This is also combined with the biogenic nature of the emissions (from renewable carbon) that the biobased bottle would release upon incineration, which do not contribute additionally to climate change.The other evaluated impacts were found to be significantly less relevant and contribute to a minor extent to the total environmental impact of PEF bottles [149].

PEF is a relatively new material and not yet commercially available, whilst the fossil-based plastic PET is a mature product (40–60 years) and is produced in a highly established process that runs close to maximum efficiency at very large scale. It is expected that the commercialization and growth of the PEF market will lead to substantial economic, technological and environmental optimizations covering the full value chain. Recent energy optimization work for a FDCA plant at industrial scale has already shown that energy consumption can significantly be reduced resulting in a further improved LCA. Furthermore, in this LCA, the current energy mix of the Netherlands was used (which still contains a low percentage of renewable energy); it is foreseen though that in the nearer future the use of both renewable heat as well as electricity will become the norm. Considering the fact that the vast majority of GHG emissions in this current version of the LCA of PEF originates from non-renewable power and heat usage in the process (production of FDCA, bottle manufacture and recycling), this foreseen change will have a very significant effect on the actual impact on the long run. It was assumed that PEF will initially end up in an open-loop recycling stream with relatively low recycling rates. Sufficient growth of the market will enable an individual material recycling stream (close-loop recycling with high efficiencies). Last but not least the PEF process will become much more efficient, both due to energy integration as well as in achieved yields. Therefore, while maturing the PEF process substantial further GHG- and other environmental benefits will be achieved when the points above have materialized. In addition, there are various other hotspots for improvement possible that may be implemented by Avantium and/or its partners within the PEF upstream or downstream value chains.

To be able to predict potential fire risks in the production and/or use of FDCA as well as PEF the fire propagation behaviour of YXY ^®^ Technology intermediates as well as PEF has been assessed and compared to other polymers [150]. PEF seems slightly better in terms of the total energy released from the combustion process than the bulk polyester PET. In addition, PEF fires result in lesser CO and soot yields compared to PET, which is proof for a better completeness of combustion [150].

### 3.5. Disruptive Technologies need Trailblazers

Disruptive technologies such as the YXY^®^ Technology need trailblazers—those who embrace new, sustainable solutions and pave the way for broader adoption. Not everyone wants to take that first step, but Avantium is. The FDCA/PEF technology truly is the first of its kind. However, development and deployment are neither quick nor easy processes. The following aspects need to be addressed before a successful market introduction can be achieved:Have a good idea;Proof of Principle: Justify the idea by R&D in the lab;Conceptual Process Design (CPD) to assess the techno economics as well as to be able to perform an *ex ante* LCA assessment. CPD is also used to target the R&D on the aspects that have the largest impacts on costs as well as sustainability;Develop and execute IP strategy, assess Freedom to Operate the technology;Proof of Concept: run the process at pilot plant scale;The technology needs to be assessed for its ability to scale;Recyclability of the anticipated materials need to be proven;The right partners along the whole value-chain need to come on board, possible from step 3 onwards;For each application, pilot and pre-marketing studies need to be conducted at relevant scale thereby needing often ton(s) of material per pilot;Address all necessary regulatory aspects for building and operating a commercial plant as well as for the products made (a.o. REACH, Food Contact (EFSA) for Europe);Deliver the foundations for large-scale manufacturing, update techno-economic as well as LCA assessments;All while testing at every stage to ensure the appropriate safety and sustainability standards are met.

Over the years, PEF has attracted the enthusiasm and support of many partners—varying from production partners (both for FDCA and PEF), offtake partners (both FDCA and PEF), brand-owners (bottles, fibres and film), governments, and financial partners. With the support of those important partners within the PEF value chain, Avantium is now ready to build and operate the world’s first FDCA Flagship Plant, meeting the growing global demand across a range of end-product markets. In combination with the plant-based feedstock, the added functionality combined with its recyclability gives PEF all the attributes required to become one of the next-generation bulk plastic materials.

## 4. How Avantium Sees the Future: On the Edge of Commercialising PEF

Avantium has taken the Final Investment Decision to build and operate the world’s first FDCA Flagship Plant, to be built in Delfzijl (The Netherlands), with construction planned to be completed by the end of 2023 and to be operational in 2024 (Avantium Press release). The rendered picture of this commercial facility is shown in Figure 13 and is set to produce 5000 tonnes of FDCA per annum (5 kta). The main focus for the FDCA produced by the Flagship Plant will be on high-value applications which can benefit from PEF’s powerful combination of sustainability and performance features. In addition to being a profit centre in its own right, the role of the Flagship Plant will be:(a)to prove the process technology at scale, and(b)to demonstrate the commercial applications of FDCA and PEF.

The business model of the FDCA Flagship Plant is based on sales of FDCA and PEF to offtake partners. In addition, it is intended to sell technology licenses to industrial collaborators who are expected to build production capacities of >100 kta based on the knowledge and experience derived from our operation of the 5 kta Flagship Plant. In parallel, Avantium will continue to work to further optimise the YXY^®^ Technology to preserve its technological advantages.

Avantium is ready to accelerate PEF’s journey towards commercial reality and its goal is to launch this technology onto the market, paving the way for global adaptation of this new polymer, and in doing so help to bring novel, sustainable packaging solutions to your supermarket and refrigerator. Avantium believes the commercialization of PEF will create long-term value for their shareholders as well as stakeholders and will also mark a significant step towards a more sustainable and circular world.

**Figure 13 polymers-14-00943-f013:**
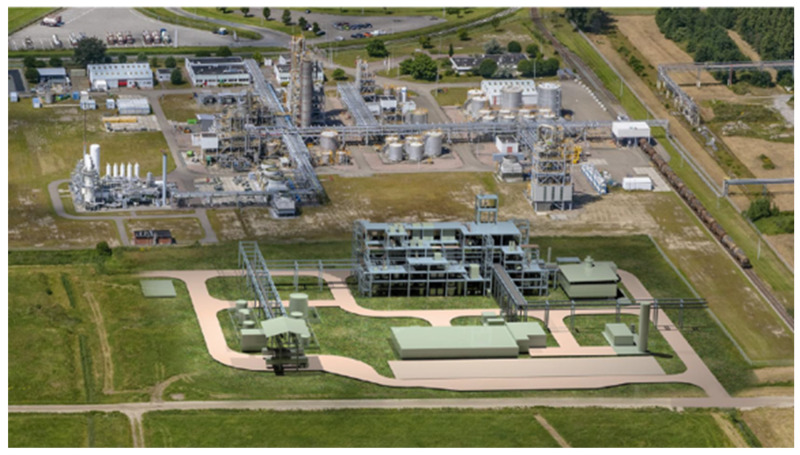
Rendered image of the Flagship Plant design of Greenfield plot at Chemie Park Delfzijl, the Netherlands.

## 5. Conclusions

As the world looks to move towards sustainable biobased polymers to replace fossil-based polymers, the challenges in doing so must be acknowledged. A new biobased polymer is by nature initially more expensive, often significantly so, than the incumbent fossil-based equivalents that have benefitted from decades of optimization and are produced at very large scale. Much of the success of bringing PEF towards commercialization has been on the basis of its excellent barrier and strength properties in comparison to the incumbent fossil-based polymers, which enables a clear justification of the higher material cost. Avantium’s trailblazer commercial partners, which are stepping forward to be early adopters of a new, more sustainable material, have been key in enabling the final steps towards commercialization. These networks of partners will be key in providing positive examples and opportunities to support the next generation of biobased polymers as they move forward in their journey to the market.

The upcoming decade will be very exciting for FDCA and PEF. Commercial volumes of PEF will enter the marketplace, hopefully creating a very positive response, new market applications and a strong drive for additional, scaled-up plants. To bring a potentially disruptive technology and a novel (bulk) polymer to the market is a herculean job and benefits enormously from an “open innovation” approach. The extensive list of joint publications with third parties is a clear reflection of the success of this approach and the establishment of PEF as the future polymer.

## Figures and Tables

**Figure 1 polymers-14-00943-f001:**
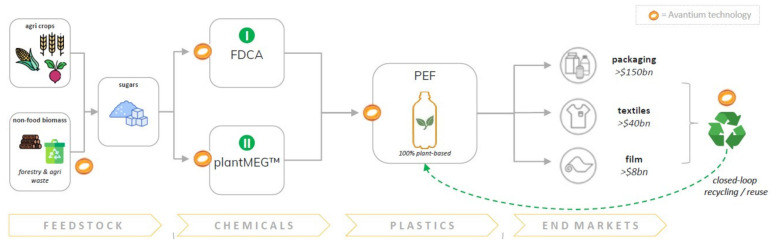
Overview of the YXY^®^ Technology value chain from feedstock towards FDCA and PEF together with the Dawn Technology^TM^ for industrial sugars, the Ray^®^ Technology^TM^ for plantMEG^TM^.

**Figure 2 polymers-14-00943-f002:**
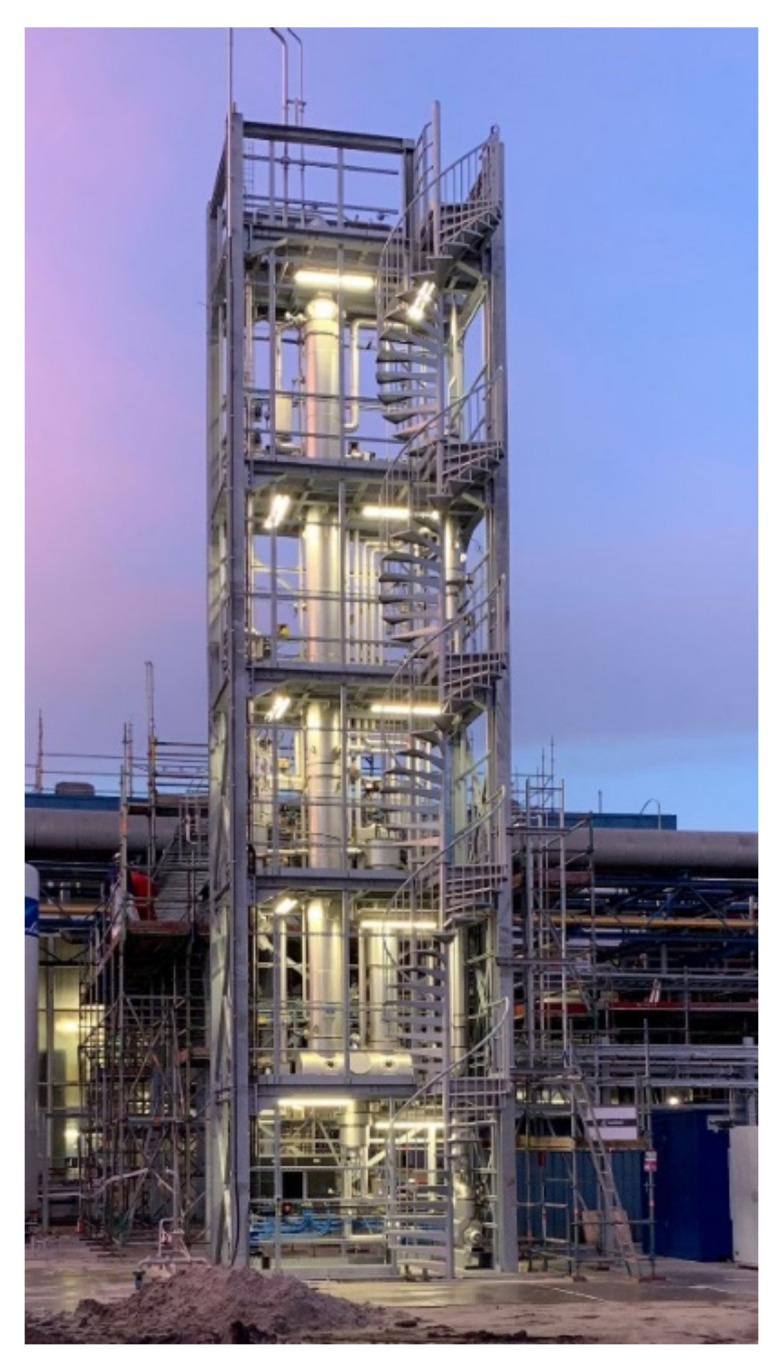
Distillation tower at Ray Technology pilot plant.

**Figure 3 polymers-14-00943-f003:**
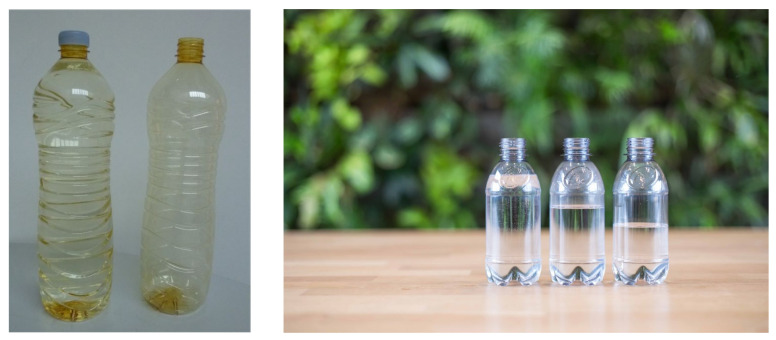
One of the first “golden” PEF bottles made (**left**) and typical examples of current fully transparent small size PEF bottles (**right**).

**Figure 5 polymers-14-00943-f005:**
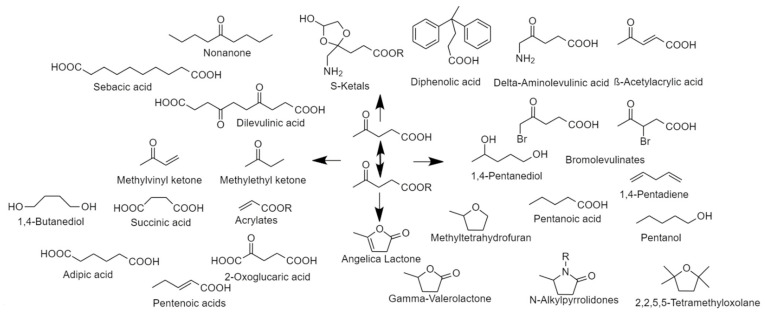
Chemical products derived from levulinic acid and its esters (adapted from [97]).

**Figure 6 polymers-14-00943-f006:**
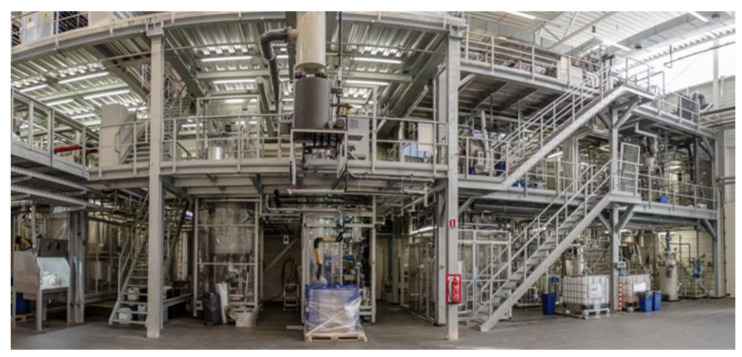
YXY^®^ Technology pilot plant in Geleen, The Netherlands consisting of an SDH unit for MMF, an oxidation unit for cFDCA and a purification unit to produce pFDCA.

**Figure 7 polymers-14-00943-f007:**
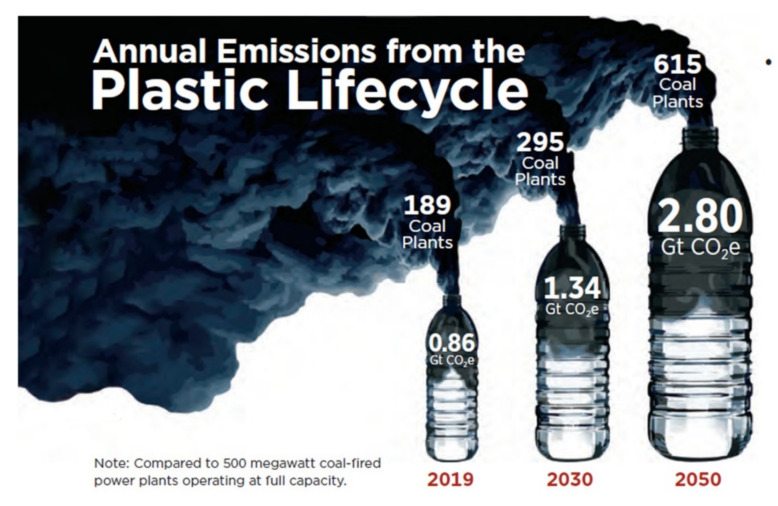
Annual emissions from the plastic lifecycle [106].

**Figure 8 polymers-14-00943-f008:**
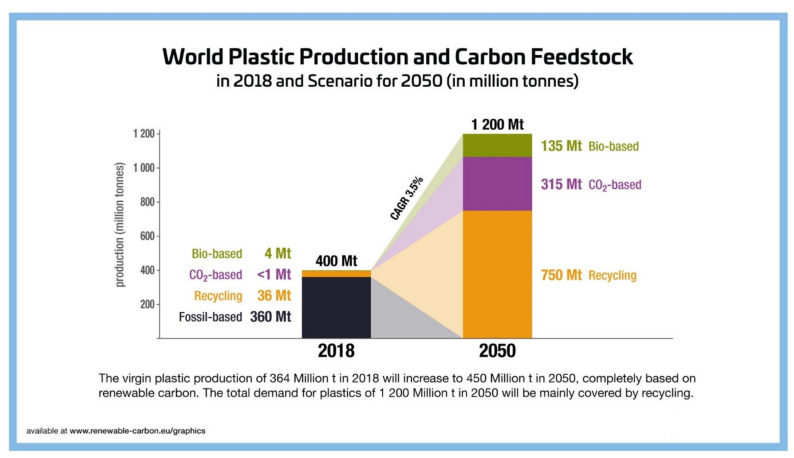
A possible scenario for the world plastic production in 2050 solely based on renewable carbon [107].

**Figure 9 polymers-14-00943-f009:**
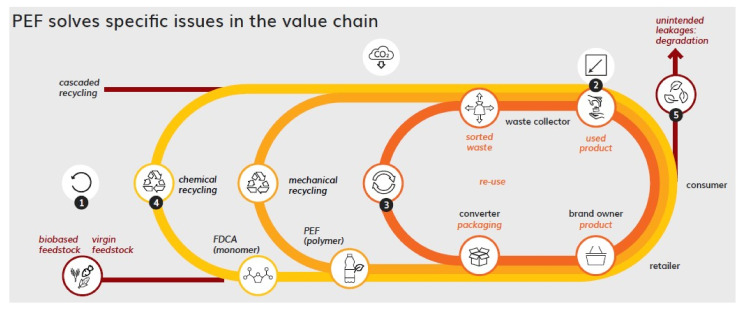
Complete overview of the circular life cycle of FDCA, PEF.

**Figure 10 polymers-14-00943-f010:**
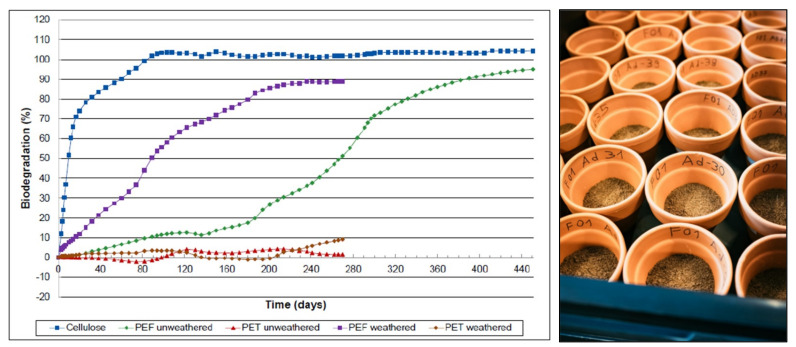
**Left**: Biodegradation profiles of weathered and unweathered PEF as well as weathered and unweathered PET and cellulose as a reference material. Biodegradation (%) = amount of polymer converted to CO_2_ (up to 450 days with air/oxygen @ 58 °C in soil. All curves are the averages of three samples. **Right**: 100′s of flowerpots containing PET films for 10-year field trials experiments in Amsterdam.

**Figure 11 polymers-14-00943-f011:**
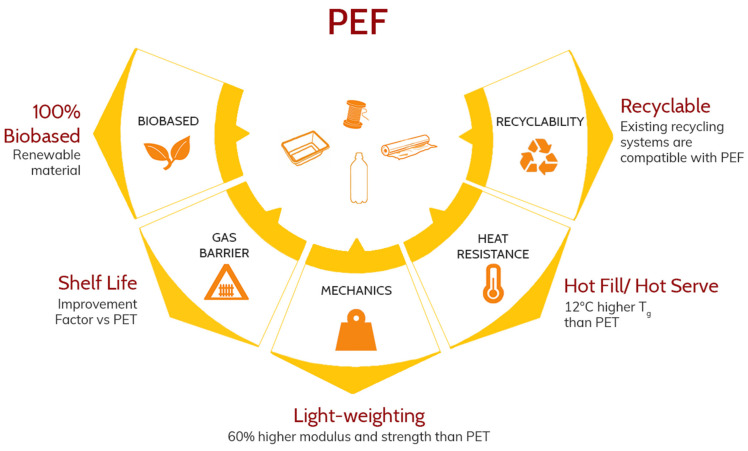
Key features of PEF compared to fossil-based alternatives such as PET.

**Figure 12 polymers-14-00943-f012:**
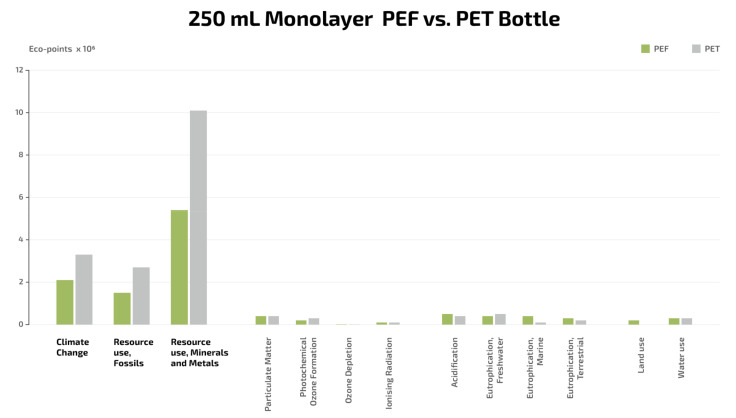
Comparative analysis of the main Life Cycle Assessment Indicators for a 250 mL PEF and PET bottle.

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
