# Peer review of "The Road to Bring FDCA and PEF to the Market"

_polymers, 2022, doi:10.3390/polym14050943_

Round 1

Reviewer 1 Report

The paper reports on the development of new bio-based polymers from initial synthesis to practical application and eventually to the market. Thus it is an unusual review: to me is more a sort of historical overview. 
Nevertheless, I found it very interesting, because it evidences the numerous problems that a brilliant idea encounter and must overcome before becoming actually realized, problems of which “laboratory researchers” – as I was – are often not even aware. The authors tell the tale of their year-long adventure with a clear, enjoyable, and convincing style. Therefore the paper deserves publication.

However, I have to make a serious objection about Figure 5. It is a pity that, when organic chemistry is involved, people tend to be unprecise and messy. As a matter of fact, in Figure 5 there are so many wrong formulae and names, that it took less time to re-draw it than to indicate the necessary corrections. 
Thus I offer you emended Figure 5 for free, as original drawing (made with ISISDraw, it can be read also with ChemDraw).

Minor remarks
-    Line 168 “to .create” : is it a full stop before create? If so, delete it.
-    Compare lines 316 and 321: either good results were always obtained on Friday afternoons only, or the year should be the same!

There are two Figure 8 (lines 594 and 633). Of course, Figures after these need to be re-numbered.

Author Response

Dear editor, reviewer,

Thank you for the review and useful suggestions for improvement. 

Thus I offer you emended Figure 5 for free, as original drawing (made with ISISDraw, it can be read also with ChemDraw). This would be very much appreciated if you redraw the figure 5.

Minor remarks
-    Line 168 “to .create” : is it a full stop before create? If so, delete it. Corrected
-    Compare lines 316 and 321: either good results were always obtained on Friday afternoons only, or the year should be the same! Corrected, should have been 2005

There are two Figure 8 (lines 594 and 633). Of course, Figures after these need to be re-numbered. Thank you, corrected

Ed de Jong

Reviewer 2 Report

The authors have succeeded in writing a useful review of advances to date in PEF manufacturing technology and the path to PEF commercialization. This review covers not only technical issues that need to be solved for PEF commercialization but also social and legal issues. This review is useful for chemists working on green polymers and managers who want to commercialize green materials.

I think this review is acceptable after making a few minor corrections.

Comment:

p.16, line 675

….by reacting with wate.

‘wate’ may be fixed to water.

p.19, line 792

(see Table 3)

‘Table 3’ is not found. It may be fixed to 'Figure 11'

Author Response

Dear Reviewer,

Thank you for your useful suggestions for improvements. We have adapted the manuscript accordingly. In more detail see our responses below:

p.16, line 675

….by reacting with wate. Thank you corrected

‘wate’ may be fixed to water.

p.19, line 792

(see Table 3) Thank you this was a mistake, reference to table 3 deleted.

‘Table 3’ is not found. It may be fixed to 'Figure 11'

Reviewer 3 Report

Dear Editor,

I have read the manuscript entitled: “The road to bring FDCA and PEF to the market” and I would like to address following suggestions to the authors

  1. In manuscript, reference numbers should be placed in square brackets [ ].
  2. The legend of Table 1 is wrong, where is PEF?
  3. The authors should check the values of glass transition temperature from ref. 137, that is reported in Table 1, line 6.
  4. Line 675, what mean authors by “reacting with wate.”
  5. Check of spelling, including:Line 168, is to . create, please delete point.; line 366, steps 1,2 and 3; line 417, etherification; line 483, tuneable properties should be tunable properties; and others.
  6. Please verify, lines 276-279, where is Table 2, is missing.
  7. Table 1 mentioned at lines 276-279, is Table 1 from Page 18?

Author Response

Dear Reviewer,

Thank you for your useful suggestions for improvements. We have adapted the manuscript accordingly. In more detail see our responses below:

  1. In manuscript, reference numbers should be placed in square brackets [ ]. Corrected
  2. The legend of Table 1 is wrong, where is PEF? Our mistake, has been corrected into PEF
  3. The authors should check the values of glass transition temperature from ref. 137, that is reported in Table 1, line 6. Corrected
  4. Line 675, what mean authors by “reacting with wate.” Corrected, should have been water
  5. Check of spelling, including: Line 168, is to . create, please delete point.; line 366, steps 1,2 and 3; line 417, etherification; line 483, tuneable properties should be tunable properties; and others. Spelling mistakes are corrected, UK English is used throughout the paper.
  6. Please verify, lines 276-279, where is Table 2, is missing. The tables refer to the Japanese regulation, clarified that content of the tables is not shown.
  7. Table 1 mentioned at lines 276-279, is Table 1 from Page 18? No it is not, see remark above under point 6.

Best regards,

Ed de Jong